# NMDA Receptor Antagonists: Emerging Insights into Molecular Mechanisms and Clinical Applications in Neurological Disorders

**DOI:** 10.3390/ph17050639

**Published:** 2024-05-15

**Authors:** Ayodeji Olatunde Egunlusi, Jacques Joubert

**Affiliations:** 1Pharmaceutical Chemistry, Faculty of Pharmacy, Rhodes University, P.O. Box 94, Makhanda 6140, South Africa; 2Pharmaceutical Chemistry, School of Pharmacy, University of the Western Cape, Private Bag X17, Bellville 7535, South Africa; jjoubert@uwc.ac.za

**Keywords:** NMDA receptor, excitotoxicity, calcium, neurodegenerative disorders, antagonists, allosteric modulator

## Abstract

Neurodegenerative disorders (NDs) include a range of chronic conditions characterized by progressive neuronal loss, leading to cognitive, motor, and behavioral impairments. Common examples include Alzheimer’s disease (AD) and Parkinson’s disease (PD). The global prevalence of NDs is on the rise, imposing significant economic and social burdens. Despite extensive research, the mechanisms underlying NDs remain incompletely understood, hampering the development of effective treatments. Excitotoxicity, particularly glutamate-mediated excitotoxicity, is a key pathological process implicated in NDs. Targeting the N-methyl-D-aspartate (NMDA) receptor, which plays a central role in excitotoxicity, holds therapeutic promise. However, challenges, such as blood–brain barrier penetration and adverse effects, such as extrapyramidal effects, have hindered the success of many NMDA receptor antagonists in clinical trials. This review explores the molecular mechanisms of NMDA receptor antagonists, emphasizing their structure, function, types, challenges, and future prospects in treating NDs. Despite extensive research on competitive and noncompetitive NMDA receptor antagonists, the quest for effective treatments still faces significant hurdles. This is partly because the same NMDA receptor that necessitates blockage under pathological conditions is also responsible for the normal physiological function of NMDA receptors. Allosteric modulation of NMDA receptors presents a potential alternative, with the GluN2B subunit emerging as a particularly attractive target due to its enrichment in presynaptic and extrasynaptic NMDA receptors, which are major contributors to excitotoxic-induced neuronal cell death. Despite their low side-effect profiles, selective GluN2B antagonists like ifenprodil and radiprodil have encountered obstacles such as poor bioavailability in clinical trials. Moreover, the selectivity of these antagonists is often relative, as they have been shown to bind to other GluN2 subunits, albeit minimally. Recent advancements in developing phenanthroic and naphthoic acid derivatives offer promise for enhanced GluN2B, GluN2A or GluN2C/GluN2D selectivity and improved pharmacodynamic properties. Additional challenges in NMDA receptor antagonist development include conflicting preclinical and clinical results, as well as the complexity of neurodegenerative disorders and poorly defined NMDA receptor subtypes. Although multifunctional agents targeting multiple degenerative processes are also being explored, clinical data are limited. Designing and developing selective GluN2B antagonists/modulators with polycyclic moieties and multitarget properties would be significant in addressing neurodegenerative disorders. However, advancements in understanding NMDA receptor structure and function, coupled with collaborative efforts in drug design, are imperative for realizing the therapeutic potential of these NMDA receptor antagonists/modulators.

## 1. Introduction

Neurodegenerative disorders (NDs) are chronic disorders characterized by the progressive loss of neuronal cells, leading to neuronal dysfunctions. These disorders are associated with a wide range of cognitive, behavioral and motor dysfunctions, including memory loss, dyskinesia, paralysis, lack of coordination, and dysphasia. These widely studied NDs, among others, include Alzheimer’s disease (AD), Parkinson’s disease (PD), Amyotrophic lateral sclerosis (ALS), Huntington’s disease (HD), Lewy body dementia, depression, and multiple sclerosis. Although they share similarities in pathology and molecular mechanisms, each ND exhibits distinct clinical, neurobiological, and pathological characteristics influenced by risk factors such as geographical variations, age, race, sex, pre-existing pathological conditions, and xenobiotics. Nevertheless, the mechanism of degeneration in each disease remains inadequately defined [1,2,3,4]. The most common forms of NDs are AD and PD. Currently, the estimated number of people worldwide suffering from dementia is 55 million. This figure is projected to rise to 78 million by 2030 and 139 million by 2050. Regarding Parkinson’s Disease (PD), approximately 3% of the global population aged over 65 years is affected [3,5]. At this rate, there is an expected substantial increase in the economic, financial, and social burden, which could have serious consequences for the overall quality of life, especially in developing countries [2,3]. In 2021, the USA spent an estimated sum of USD 355 billion and USD 52 billion on dementia and PD, respectively. Globally, the cost associated with AD is approximated at USD 1 trillion annually, and this amount is projected to increase in the future due to the high ageing population [6,7].

Over the years, numerous studies have identified molecular and cellular mechanisms, giving rise to several etiological hypotheses. However, the mechanism of degeneration is still poorly defined. Despite many therapeutic trials derived from these hypotheses, none has been successful, as current treatments only offer symptomatic relief without halting the degenerative process or regenerating the neurons [8]. The challenge could be attributed to the multifactorial nature of these NDs, as each disorder is a result of interrelated processes that include oxidative stress, excitotoxicity, neuroinflammation, genetic mutation, endoplasmic reticulum dysfunction, protein aggregation and mitochondrial dysfunction [9,10]. Since the majority of NDs are sporadic, excitotoxicity is prominent among the proposed degenerative mechanisms. Despite the development of several molecules to address some of these mechanisms of degeneration, many have failed primarily due to their inability to cross the blood–brain barrier (BBB), a tightly spaced network of blood arteries and endothelial cells that makes ND treatments extremely complex and challenging [11,12]. 

Glutamate, the most vital excitatory neurotransmitter in the central nervous system (CNS), plays a crucial role in regulating various metabolic pathways. Under physiological conditions, the concentration of glutamate within the synapse is carefully regulated and maintained through neuron–astrocyte interaction, ensuring a physiological concentration in the extracellular space [13,14]. This glutamate homeostasis together with ion homeostasis is essential for preserving normal glutaminergic brain functions, including synaptic formation and signaling, neuronal plasticity, neurotransmission, learning, memory, and ageing. However, in a diseased state, this homeostatic balance is compromised, leading to an increase in glutaminergic neurotransmission and dysfunction resulting in excitotoxicity. In excitotoxicity, excessive extracellular glutamate overactivates the *N*-methyl-D-aspartate (NMDA) receptor, causing a significant intracellular calcium overload. This overload triggers a cascade of events that eventually leads to neuronal cell death either by apoptosis or necrosis (Figure 1) [15,16,17,18,19,20]. This form of neuronal death occurs gradually over a long period and has been implicated in the physiopathology of the most common NDs, including AD, PD, ALS and HD, especially in their early phases [21]. As such, minimizing glutamate activity, either through the synaptic clearing of excess glutamate or modulating NMDA receptors, could be therapeutically beneficial in addressing excitotoxic-mediated neuronal cell death. While the former exists under physiological conditions through astrocyte–neuron interactions, it becomes compromised in the diseased state, as observed in most NDs. This makes antagonizing the NMDA receptor a compelling strategy in slowing or halting the degenerative process and relieving symptoms associated with NDs, especially where excitotoxic-mediated death is concerned. Several NMDA receptor antagonists have been explored, but only a few, including amantadine for PD, memantine for AD, and riluzole for ALS, have been successful in clinical trials. However, these successes are not without drawbacks, as they are associated with several side effects that hinder their adherence. Moreover, many NMDA receptor antagonists have failed in clinical trials due to undesirable extrapyramidal effects and pharmacokinetic challenges. 

The review aims to comprehensively explore the molecular mechanism underlying NMDA receptor antagonists at their respective binding sites. Before delving into this, we will offer insight into the structure and functions of NMDA receptors. We will also categorize these antagonists/modulators based on their binding sites and highlight the associated side-effect profiles, particularly those that have impeded their development. Furthermore, we will propose new directions or avenues for investigating NMDA receptor antagonists as potential treatments for NDs. Understanding these concepts could pave the way for innovative strategies in the treatment of neurodegenerative disorders. Additionally, current challenges and future perspectives in the field of NMDA receptor antagonists will also be discussed.

## 2. Structure and Functions of NMDA Receptors

The NMDA receptor is one of the ionotropic glutamate receptors that carries out excitatory neurotransmission in the CNS [22]. Under resting conditions, the NMDA receptor, primarily located in the postsynaptic site of neurons, is blocked by Mg^2+^. However, upon activation by glutamate or postsynaptic depolarization, it becomes highly permeable to cations, predominantly calcium ions. The NMDA receptor is divided into three subunits: GluN1, GluN2 and GluN3 subunits. The GluN2 subunits are further divided into four subtypes (GluN2 A-D), while GluN3 is subdivided into two types (GluN3 A-B). Despite the distinct biochemical and biophysical properties exhibited by GluN1 and GluN2 subunits, their combination forms the traditional heterotetrameric NMDA receptor. The resulting complex or channel is composed of one or more of the GluN2 subtypes coupled with the GluN1 subtype. This receptor, upon binding with glycine or D-serine (co-agonists) and glutamate (agonist), respectively, is necessary for optimal NMDA receptor functions or activation. This dual agonism on the NMDA receptor for its cellular function, a distinct feature that sets it apart from other neurotransmitter receptors, remains a topic of debate. However, the consensus is that glutamate is responsible for triggering NMDA receptor activation, and glycine or serine is essential for controlling the level of receptor activity. While the presynaptic function of NMDA receptors mediates neurotransmitter release and long-term plasticity, activities at the postsynaptic part of neurons are responsible for its slow current and synaptic plasticity. The binding affinity of glycine or D-serine to the GluN1 subtype of NMDA receptor is contingent on the specific brain region [7,16,23,24,25,26,27,28,29,30]. In contrast, the NMDA receptor channel formed by GluN1/GluN3 subunits is less sensitive to Ca^2+^ influx, not readily influenced by Mg^2+^ block, and can only be activated by glycine alone [28]. Thus, this type of NMDA receptor complex is less involved in Ca^2+^-medicated responses and is likely to have minimal impact on excitotoxicity.

Interestingly, GluN1 and GluN2 subunits share a fundamental structural similarity with other glutamate-gated ion channels or ionotropic glutamate receptors. All ionotropic glutamate receptor structures are classified into domains, and each subunit polypeptide chain consists of an amino-terminal domain (ATD), a ligand-binding domain (LBD), a transmembrane domain (TMD), and a carboxy-terminal domain (CTD). While the ATD is mainly responsible for the assembly and regulation of subunits, CTD plays a major role in receptor transport and anchoring the receptor to other intracellular molecules, enabling optimal interaction. However, the distinction lies in the presence of asparagine residue within the second transmembrane domain (M2 loop) of the GluN1 and GluN2 subunits. This region is suggested to serve as the pore-forming part of the NMDA receptor subunit and may be responsible for the ion permeability of the channel. Other transmembranes associated with the ionotropic structure are three membrane-spanning helices identified as M1, M3, and M4 [23,24,28,30,31,32,33,34]. Another distinctive feature between the NMDA receptor and other ionotropic receptors is the proximity of GluN2-ATD to the LBD of a GluN1/GluN2C receptor complex, enabling the binding of a positive allosteric modulator [34]. In terms of localization in the CNS, the GluN1 subunit is expressed ubiquitously at every developmental stage. However, this is not the case for the GluN2 subunit, as some subtypes exhibit uneven distribution, especially in the adult stage. While the GluN2A subtype is expressed widely, GluN2B, GluN2C and GluN2D are predominantly expressed in the forebrain (cortex, striatum, and hippocampus), cerebellum and midbrain, respectively. With this differential distribution, there is a possibility of targeting specific GluN2 subtypes with scaffolds that address a particular neurodegenerative disease and exhibit a good side-effect profile [7,31,33,35]. However, GluN2A and GluN2B are the main functional ion channels in the CNS. This is due to the low probability of channel opening in GluN2C and GluN2D subtype receptors [36]. Therefore, selectively targeting GluN2A or GluN2B subunits could be significantly impactful in the development of potential therapeutical agents. However, one needs to be mindful of the potential adverse effects posed by these agents. 

As illustrated in Figure 2, the binding of glutamate and/or glycine to the LBD of GluN2 or GluN1 subunits, respectively, which occurs during the transition of the NMDA receptor from a resting state to an active state, causes the LBD bi-lobe to close and subsequently pulls the LBD-TMB linkers. This results in the opening of the ion channel pore, leading to the influx of Ca^2+^ [28,30]. This NMDA receptor-mediated Ca^2+^ response is responsible for mediating long-term potentiation and synaptic plasticity, the cellular basis of learning and memory, and maintaining neuronal health. The physiological functions of NMDA receptors are determined by their subunit composition, the location of subunits within the CNS, and the developmental stages of the brain (from embryo to adult) [26,34]. Noteworthy, recent studies in mice have observed a reduction in the expression and functions of the NMDA receptor, and the diffusion of the NMDA receptor, especially the GluN2B subtype, to the dendritic spine, leading to the formation of extrasynapse. This phenomenon is associated with advanced ageing, and the activation of these extrasynaptic NMDA receptors is implicated in neuronal cell death and accelerated age-related cognitive decline [37]. As such, targeting the NMDA receptor has been suggested to be therapeutically relevant and useful in addressing various neurodegenerative disorders. So far, a great deal of these targets have been developed and explored [28].

## 3. Types and Molecular Mechanisms of NMDA Receptor Antagonists

Several studies have implicated excitotoxicity as a prominent factor in the pathogenesis of most neurodegenerative disorders. This makes exploring NMDA receptor antagonists a potentially viable therapeutic approach to addressing these disorders. A number of these NMDA antagonists have been explored and investigated as potential neuroprotective agents. Despite the extensive research in this area, there has been an uphill challenge in developing these antagonists into effective therapeutical tools due to the compromised physiological function of NMDA receptors, resulting in extrapyramidal side effects such as cognitive impairment, hallucination, and psychosis. Nevertheless, the distinct subunits feature in different parts of the brain, particularly the GluN2 subunit, offering hope for the design and development of subunit-type antagonists with acceptable side-effect profiles [38]. These antagonists are categorized based on their binding sites into competitive, non-competitive, and negative allosteric antagonists, and their NMDA receptor subunits/subtypes, pharmacological and side-effect profiles are highlighted in Table 1.

### 3.1. Competitive NMDA Receptor Antagonist 

Understanding the molecular mechanism underlying neurodegenerative disorders led to the development of competitive NMDA receptor antagonists (Figure 3) such as D-CPP (D(-)3-(2-carboxypiperazin-4-yl)-propyl-l-phosphonic acid), D-CPP-ene (D(-)3-(2-carboxypiperazin-4-yl)-propenyl-l-phosphonic acid; Midafotel) D-AP5 (2-amino-5-phosphonopentanoic acid, D-AP7 (2-amino-7-phosphonoheptanoic acid), DCKA (5,7-dichlorokynurenic acid), CGP-78608 (2-amino-4-methyl-5-phosphono-3-pentenoate-l-ethyl ester, CGP-37849 (DL-(E)-2-amino-4-methyl-5-phosphono-3-pentenoic acid), CGS-19755 (l-cis-2-carboxypiperidine-4-yl)-propyl-l-phosphonic acid; Selfotel), SDZ-220-040 ((s)-α-amino-2,4,-dichloro-4-hydroxy-5-(phosphonomethyl)-[1,1-biphenyl]-3-propanoic acid), L689-560 (trans-2-carboxy-5,7-dichloro-4-phenylaminocarbonylamino-1,2,3,4-tetrahydroquinoline), L-701,324 (7-chloro-4-hydroxy-3(3-phenoxy)phenyl-2(H)quinoline), PPDA (2S, 3R)-1-(phenanthrene-2- carbonyl)p iperazine-2,3-dicarboxylic acid, NVP-AAMO77 ([[[(1S)-1-(4-bromophenyl)ethyl]amino] (1,2,3,4-tetrahydro-2,3-dioxo-5-quinoxalinyl)methyl]phosphonic acid tetrasodium salt; PEAQX) and ST3 (((S)-5-[(R)-2-amino-2-carboxyethyl]-1-[4-(3-fluoropropyl)phenyl]-4,5-dihydro-1H-pyrazole-3-carboxylic acid). These antagonists bind directly to the binding sites of glycine or glutamate at the GluN1 or GluN2 subunits, respectively. To understand the binding interaction of these antagonists, analysis of the cryo-EM structure of an intact NMDA receptor complex bound to antagonists or FRET analysis of the crystal structure of the LBD indicates an increase in the opening of the GluN1 and/or GluN2 clamshell by various degrees (13–28°) when compared to glycine or glutamate binding, respectively, occupying the same active binding site. Subsequently, these clamshell openings lead to the relaxation of the tension in the LBD-TMD linker, resulting in the closure of the ion channel pore (Figure 4) [28,39,40,41]. 

Functional NMDA receptor antagonists have been shown to exhibit anticonvulsant, anti-ischemic, antidepressant-like and anxiolytic-like properties [39,42,43]. Despite their strong activities in attenuating glutamate-medicated excitotoxicity, these antagonists have been marked by unfavorable side effects such as hallucination, agitation, confusion, paranoia, delirium, drowsiness, and coma. These adverse effects render them unsafe for human use, leading to their failure in clinical trials [44,45]. A notable example is D-CPP-ene, initially touted as a promising antiepileptic agent due to the absence of phencyclidine-like adverse effects observed at therapeutic doses in pre-clinical studies. It was also well tolerated in phase I clinical trials, with healthy volunteers exhibiting tolerance up to a dose of 2000 mg/day. However, D-CPP-ene was terminated at phase II due to severe adverse effects, including confusion, disorientation, gait ataxia and sedation, or worsened seizures noted at daily doses of 500–1000 mg/day [42]. Moreover, the majority of competitive NMDA antagonists except SDZ 220-581 permeate poorly across the BBB due to their hydrophilic nature [44,46,47]. Table 1 illustrates the pharmacological action and adverse effects associated with a few competitive NMDA antagonists that were investigated for neurological disorders. Despite a decade-long search for a competitive antagonist with a good safety profile and minimal side effects, none has completed clinical trials due to the associated psychotomimetic or dopaminergic transmission side effects. Generally, antagonists targeting GluN2 (A-D) are more prone to these unwanted adverse effects than the GluN1 subunits. While several non-selective NMDA or GluN2 antagonists have displayed psychotomimetic and/or dopaminergic side effects similar to MK-801, GluN1 antagonists have shown more favorable outcomes [48,49,50,51]. Therefore, it is suggested that GluN1 antagonists could potentially address various neurological disorders. However, available information is derived only from preclinical studies on disorders such as anxiety, depression, and epilepsy. Moreover, the clinical relevance of a GluN1 antagonist is highly debatable because of the wide distribution of GluN1 subunits in the central nervous system or the important functions of co-agonists in NMDA receptor physiological functions. Clinical studies for this class of antagonists are needed to confirm their therapeutic relevance. Noteworthy is the limited preclinical data for complex neurodegenerative disorders like AD and PD [28,39,41,42,46,52,53,54,55,56,57,58,59,60]. Preclinical and clinical studies are essential to bridge this knowledge gap and determine the potential efficacy of GluN1 antagonists in treating these neurodegenerative disorders.

### 3.2. Uncompetitive or Non-Competitive NMDA Receptor Antagonists

Decades ago, the failure of competitive NMDA receptors to effectively address neurological disorders clinically redirected research towards non-competitive NMDA receptor antagonists. During this period, the focus was on targeting this binding site to tackle neurodegenerative disorders such as depression [61], PD, and AD. The aim was to alleviate the undesirable adverse effects associated with competitive NMDA receptor antagonists. These non-competitive NMDA receptor antagonists, also known as channel blockers, include phencyclidine (PCP), dizocilpine maleate (MK-801), ketamine, and tiletamine (Figure 5). They act by binding with high affinity to the PCP binding site at the entrance of the channel gate, as shown in Figure 6, to block calcium-mediated responses. Consequently, they have demonstrated neuroprotective effects in conditions such as stroke, cardiac arrest, and neurodegenerative disorders. Moreover, they have been shown to display anti-dyskinetic and anticonvulsant effects, although variations may arise depending on the rodent strains or models employed [62,63,64,65,66,67]. These variations have led to ambiguous effects in the literature data on some of these open-channel blockers. For instance, one group found no anticonvulsant effects with ketamine in a 4-aminopyridine (4-AP) induced epileptic model of hippocampal slices, while another group was able to indicate its anticonvulsant properties in a 4-AP-induced seizure model of male Wistar rats [62]. The differences in effects could be attributed to the variation in NMDA receptor subunit complexes expressed by the cells or animals, influenced by age. 

Despite their efficacy, due to their high affinity, they bind rapidly and dissociate slowly, prolonging the calcium-mediated response and resulting in unfavorable clinical outcomes [68]. Similar to most competitive NMDA receptor antagonists, they are known to induce adverse effects, such as neuropsychological, psychotomimetic, and dopaminergic transmission effects, which limit their clinical use. Some of these antagonists can induce schizophrenia-like symptoms, even in healthy volunteers. The dopaminergic transmission effects of non-competitive NMDA receptors stem from their ability to activate the dopaminergic system, subsequently increasing dopamine synthesis, release and metabolism in various parts of the brain [64,68,69]. Confounding the problem is the influence of these antagonists on the increase in turnover and release of serotonin, which are known to exacerbate schizophrenia-like symptoms [70].

The rediscovery of clinically tolerated memantine and amantadine as non-competitive NMDA receptor antagonists marked a significant breakthrough in the treatment of neurodegenerative disorders. Much like MK-801, both drugs bind to the PCP binding site of the ion channel complex. However, the efficacy and clinical use of amantadine and memantine as anti-dyskinesia (in PD) and a neuroprotective agent (in AD), respectively, are partially dependent on its weak NMDA receptor antagonist [66,71]. This clinical tolerability is attributed to the considerably shorter residence time within the channel in an open state when compared to MK-801. Interestingly, a study demonstrated the ineffectiveness of memantine at low-level NMDA receptor activation but found it to be highly efficacious in the overactivation state. This favorable kinetics makes these blockers better neuroprotective agents with minimal side-effect profiles when compared to MK-801 [71,72,73,74,75]. Additionally, both amantadine and memantine are known to attenuate epileptiform activity behaviors induced by 4-AP in a rat model. When compared to amantadine, memantine has displayed better therapeutical indices in the management of epilepsy and other neurological or psychological disorders [62].

Similar to competitive NMDA receptor antagonists, some non-competitive antagonists have been associated with undesirable adverse effects. These effects are not only attributed to their strong binding affinity to the PCP binding site but also to their promiscuity. For example, antagonists like ketamine are known to bind to other receptors, enhancing the activity or transmission of other neurotransmitters such as dopamine, serotonin, noradrenaline α-Amino-3-hydroxy-5-methyl-4-isoxazolepropionic acid (AMPA), acetylcholine, opioid, and gamma-aminobutyric acid (GABA) [76,77,78]. This can lead to unwanted side effects, including addiction, dependency, and tolerance, particularly with prolonged use.

### 3.3. Allosteric NMDA Receptor Antagonists

Despite numerous studies on competitive and uncompetitive NMDA receptor antagonists, addressing neurodegenerative disorders still poses enormous challenges, as current treatments only offer symptomatic relief. To date, no drug has successfully halted or slowed down the degenerative process. An alternative to competitive and noncompetitive antagonists is the allosteric modulation of NMDA receptors. The influence of GluN2 on the biophysical characteristics of NMDA channels provides a potential allosteric target [79,80,81]. For instance, the GluN2B subunit presents an attractive site for therapeutic interventions in chronic neurodegenerative diseases like PD, AD, ALS, HD, and multiple sclerosis, as well as in acute neuronal diseases like traumatic brain injury, epilepsy, and stroke. What makes this target particularly intriguing is its enrichment in perisynaptic and extrasynaptic NMDA receptors, major contributors to excitotoxic-induced neuronal cell death. However, antagonists targeting this subunit have failed in clinical trials due to their lack of benefit in PD patients. Like non-competitive antagonists, the binding of allosteric NMDA antagonists to their receptor is independent of the presence or absence of agonists (glutamate, glycine or serine) at the binding site [27,82,83,84,85,86,87,88,89,90,91,92,93,94]. Allosteric modulators (Figure 7) of NMDA receptor channels can be classified into positive and negative modulators, and several of them have been explored extensively. Both positive and negative modulators bind to the ATD part of the NMDA receptor channel to potentiate or block, respectively, Ca^2+^-mediated responses. As shown in Figure 8, modulators like ifenprodil, a selective GluN2B inhibitor, act by binding to the interface between the GluN1 and GluN2B ATDs, and the mobility of the GluN2 lower lobe is vital for NMDA receptor inhibition [34,80,88,89,95,96,97,98]. This was confirmed by a cross-linking study that indicated a decrease in distance between the lobes of GluN2 ATD for the ifenprodil-bound NMDA receptor and immobile GluN (1 and 2) ATD upper lobes, despite conformation changes within the ATDs [80]. Similar to ifenprodil, are radiprodil and Ro 25-6918, which have been shown to selectively inhibit GluN2 subunits to antagonize NMDA receptors. These selective GluN2B antagonists are efficacious with reduced side effects against a few neurodegenerative disorders. However, clinical trials for ifenprodil and radiprodil were terminated early due to poor bioavailability and lack of recruitment, respectively. For instance, radiprodil was initially investigated for infantile spasm syndrome due to its stronger anticonvulsant effects in younger rat pups compared to adult animals. However, it was terminated in the early stage of phase 2 clinical trials because of challenges in recruiting infant patients within the prescribed timeframe [89,99,100]. Ifenprodil was investigated as an adjunct therapy in PD patients with waning efficacy of levodopa but failed in phase 2 clinical trials as the drug failed to reduce tremor, rigidity and bradykinesia due to poor BBB permeability [101]. However, it found success in treating cerebral ischemic disease and is currently marketed in Japan and France as Cerocral^®^, a cerebral vasodilator [102]. 

Other negative allosteric modulators include DQP-1105 which has been shown to selectively block the GluN2D subtype to regulate synaptic transmission in the subthalamic nucleus, substantia nigra, striatum and spinal cord, and the GluN2C to presynaptically modulate gabaminergic synaptic transmission in the suprachiasmatic nucleus [103]. TCN-201 (sulfonamide derivative) was also identified as a promising selective GluN2A allosteric modulator but could not proceed to biological studies due to its poor solubility [27,82,96,104]. However, analogues of TCN-201 like MPX-004 and MPX-007 with enhanced solubility and increased efficacy have been designed and explored. These analogues could provide opportunities to further understand the mechanism of GluN2A NMDA receptor allosteric modulators [27,30]. 

Regrettably, the majority of active drugs targeting allosteric sites have not produced the desired therapeutic success in clinical trials. However, recent developments have identified a series of phenanthroic and naphthoic acid derivatives with enhanced GluN2B, GluN2A or GluN2C/GluN2D selectivity, depending on their functional moieties, and improved pharmacodynamic properties [92,96,104,105,106]. Additionally, these derivatives are amphipathic, possessing both hydrophobic and charged moieties. This characteristic promotes BBB permeation via the neurosteroid transporter, suggesting a good bioavailability for this group of compounds, particularly in addressing NDs [92,96]. 

**Table 1 pharmaceuticals-17-00639-t001:** Pharmacological and side-effect profiles of developed NMDA receptor antagonists.

Antagonist Type	Compounds	Receptor Subunits/Subtypes	Developmental Stage	Pharmacological Profiles	Side-Effect Profiles	References
Competitive antagonists	D-CPP/D-CPP-ene (Midafotel)	GluN2A	Terminated at Phase 11 clinical trials	Antiepileptic and neuroprotective effects against head injury, cerebral ischemia and stroke Alteration of acute behavioural response to cocaine. Stimulate short-term increase in NREM (non-rapid eye movement) sleep	Hallucinations Poor concentration Confusion Gait ataxia Sedation Depression	[28,39,42,54,55,68,107]
	D-AP5/D-AP7	Non-subunit selective	Preclinical or experimental studies	Block fear acquisition and expression Block or interfere with acute response to psychostimulants such as cocaine amphetamine, or methamphetamine	Similar to those of D-CPP-ene/D-CPP	[28,55]
	DCKA	GluN1	Preclinical or experimental studies	Anxiolytic effect Neuroprotective against NMDA/glycine-induced toxicity	Lack of psychotomimetic effects or side effects associated with dopaminergic transmission	[28,56,108,109,110]
	CGP-78608, CGP-37849 & CGP-40116	GluN1	Preclinical or Experimental studies	Anticonvulsant effect	Lack of side effects associated with dopaminergic transmission	[28,45,64,68,111]
	CGS-19755 (Selfotel)	GluN2A	Terminated at Phase III clinical trials	Neuroprotective effect against global and focal ischemia, trauma and stroke	Psychotomimetic side effects like Hallucination Confusion Paranoia Delirium Lack of side effects associated with dopaminergic transmission	[28,44,46,52,53]
	L689-560 & L701-324	GluN1	Preclinical or experimental studies	Anticonvulsant effects Anxiolytic Antidepressant-like effect in mice	Sedation Lack of neuronal vacuolisation and psychotomimetic potential Ataxia at a high dose Modest impairment of reference memory, but no negative effect on working memory	[28,49,56,112,113,114,115,116]
	PPDA	GluN2A, GluN2C &GluN2D	Preclinical or experimental studies	Prevent the complete worsening effect of tissue-type plasminogen activator on NMDA-induced neuronal death in both cultured cortical and hippocampal neurons Anti-allynic and anti-hyperalgesic effects in rat	Motor dysfunction at high dose	[28,41,58,60,117,118,119,120,121]
	NVP-AAMO77 (PEAQX)	GluN2A GluN2C & GluN2B	Preclinical or experimental studies	Produce anti-compulsive behaviour in a rat model Impairment of contextual and temporal fear responses Antidepressant-like effect in rodents	Affect motor coordination stamina and motivation run in a rat dyskinesia model Motor memory impairment or learning memory deficit	[28,36,117,119,122,123,124,125,126]
	SDZ-220-040	GluN2B	Preclinical or experimental studies	Design to readily cross the BBB. Effectively disrupt prepulse inhibition in rats Anticonvulsant effect Protection against focal ischemia Attenuate neuropathic pain	Sedation Ataxia Psychotomimetic effects	[28,127,128,129]
Non-competitive antagonist	MK-801	Open-Channel blocker	Preclinical or experimental studies	Reverse mild stress-induced anhedonia in male Wistar rats Neuroprotective effect in several animal models of cerebral ischaemia Block L-Dopa-induced dyskinesia in a rat preclinical model, but only at concentrations that worsen parkinsonism Anti-convulsant effect	Weight loss Hypothermia Death Hallucination Ataxia Hyperlocomotion	[53,61,63,66,130,131]
	Memantine	Open-channel blocker	Approved for AD in human	Neuroprotective effect in AD, vascular dementia and prodromal stages of psychosis Antidepressant-like effect Antinociceptive effect in rats Anticonvulsant effect	Occasional restlessness Headache Hypertension Drowsiness Constipation Diarrhoea Nausea Anorexia Dyspnea Slight dizziness at a high dose	[45,62,71,72,75,132,133,134]
	Amantadine	Open-channel blocker	Approved for PD in human	Anti-dyskinetic effect Effectively reduce L-Dopa-induced abnormal involuntary movement Anti-convulsant effect Neuroprotective effect	Visual hallucination Confusion Blurred vision Leg oedema Dry mouth Constipation Urinary retention	[62,63,66,131,135]
	PCP	Open-channel blocker	Preclinical/experimental studies	Anticonvulsant effects in NMDA- or quinolate-induced seizure model Anaesthetic effect	Hallucination Ataxia Hyperlocomotion Emergency delirium	[45,136]
	Ketamine	Open-channel blocker	Approved as an anaesthetic agent	Anticonvulsant effect in NMDA- or quinolate-induced seizure model Anaesthetic effect Antidepressant effect in resistant major depressive disorder	Induce cognitive deficits and psychotic symptoms Hallucination Abuse Psychological and physiological dependences Possible neurotoxicity Nystagmus Drowsiness Nausea and vomiting Blood pressure elevation Liver and bladder damage	[45,130,132,133,136,137,138,139]
	Tiletamine	Channel blocker	Approved for veterinary use	Anaesthetic effect Anticonvulsant effects in NMDA- or quinolate-induced seizure model	Robust Sedation in human and animal Ataxia Feeling of dissociation Hallucination	[136]
Negative allosteric modulator	Ifenprodil	GluN2B	Phase III clinical trials completed	Neuroprotective effect in both in vitro and in vivo models of cerebral ischemia Anticonvulsant effects in rodent Rapid antidepressant effect Alleviate neuropathic pain	Impair cognitive behavioural tasks	[28,81,85,93,100,134,140,141,142]
	Radiprodil	GluN2B	Terminated at Phase II clinical trials	Anticonvulsant effect in rate model (stronger in young rat pups than adult animals) Decrease epileptic spasms in infants	Vomiting Pyrexia	[100]
	Ro25-6981	GluN2B	Preclinical or experimental studies	Rapid antidepressant effect and counteract depressive-like behaviour in chronically stressed rodent Neuroprotective effect against glutamate-induced toxicity in a cultured cortical neuron Improve anxiety and compulsive behaviour in obsessive-compulsive disorder rat Alleviate cerebral ischemia-reperfusion and oxidative damage in male Sprague Dawley rats Antipakinsonian effect in 6-OHDA-lesioned and MPTP PD rat model	Reduced memory in early life stress mice	[28,88,143,144,145,146]
	DQP-1105	GluN2C & GluN2D	Preclinical or experimental studies	Neuroprotective effects in GluN2D-rich substantia nigra compacta dopaminergic neurons	Motor dysfunction	[28,103,147,148,149,150,151]

## 4. Current Challenges and Future Perspectives

Over the years, translating preclinical studies of NMDA receptor antagonists into successful clinical drugs has been an uphill task. This is mainly due to the complexity of the NMDA receptors, as their optimum function is needed for several brain physiological functions. Deviation in the form of hypofunction or hyperfunction is detrimental to general well-being [30]. Despite decades of studies, the use of NMDA receptor antagonists to address these defects in many neurodegenerative disorders while still maintaining the optimum physiological function of the NMDA receptor has yet to produce the desired outcome. Not only are there conflicting results in preclinical studies, but therapeutic effects observed in animal studies have failed to translate into human studies. This suggests a limited understanding of the NMDA receptor’s physiological functions in human brains. Confounding this issue is the presence of the heterodimeric (GluN1/GluN2B or GluN1/GluN2D) heterotrimeric (GluN1/GluN2B/GluN2D) structure of NMDA receptors in certain neurons, which significantly increases the difficulty in resolving the function of different NMDA receptor subtypes [96,150]. Moreover, several of these antagonists are marked by undesirable adverse effects like psychotomimetic, dopaminergic transmission, or schizophrenia-like effects. These adverse effects are linked to either the competitive inhibition of endogenous excitatory neurotransmitters, the strong binding affinities of these antagonists, the activities at other neurotransmitter receptors, or the inability of NMDA receptors to metamodulate other neurotransmitter receptors [25]. Even though clinically tolerated NMDA antagonists (memantine and amantadine) are available, they are not without side effects. Interestingly, a few selective allosteric modulators (ifenprodil) targeting NMDA receptors have shown promising results with good side-effect profiles when compared to competitive and noncompetitive NMDA receptor antagonists. However, they are often poorly bioavailable, and their selectivity to a particular NMDA receptor subtype or subunit is relative. Not only are they difficult to study despite being knowledgeable about the binding sites [88], but they are also known to interact with other neurotransmitter receptors. For instance, ifendropil, a selective GluN2B negative allosteric modulator, has been shown to also act at adrenoceptor 1 & 2 (α_1&2_), sigma and (serotonin) 5-TH_1A&2_ receptors [85,90,152]. Similar GluN2B antagonists. like Ro-25698, with a low crossreactivity with adrenoceptors, are known to bind to 5-HT, histamine-1 (H_1_) and sigma receptors [152]. Thus, they can cause unwanted side effects. Nevertheless, the significance of the GluN2B subunit, particularly those at the extrasynaptic sites, in excitotoxic-induced cell death makes them a favorable therapeutic target for any potential modulators. Modulators with enhanced solubility and bioavailability must be specific and selective to the GluN2B subtype. To achieve this, a better understanding of the mechanism of allosteric binding of antagonists or modulators to NMDA subunits is needed, but remains poorly defined to date. 

To improve the solubility and bioavailability of these GluN2B antagonists, researchers could explore nano-drug formulations, focusing on solid lipid nanoparticles. This form of drug delivery system would involve incorporating the drug molecules in stearic acid/poloxamer188 nanoparticles and coating them with chitosan. These nanoparticles, with an average size of 300 nm, through intranasal administration, have been shown to effectively cross the BBB in brain endothelial cell permeation and uptake studies. Moreover, the cationic nature of chitosan may aid adhesion to brain endothelial cells, improving the transport of nanoparticles. Additionally, the nose–brain delivery system provides an enriched blood vessel that enhances the bioavailabilities of loaded drugs. Similar drug delivery systems have been used to improve the BBB permeability of dopamine and riluzole, with the potential to treat PD and ALS, respectively [153,154,155]. Surprisingly, to date, no studies have explored or reported on this form of drug delivery system for ifenprodil and its derivatives, which were once touted as promising drug agents for the treatment of NDs. Exploring such an avenue could provide the much-needed breakthrough for this class of antagonists in halting or slowing the degenerative processes mediated by glutamate-induced toxicity.

Recently, fluoroethylnormemantine (FNM), a novel NMDA receptor antagonist derived from memantine, has been developed for the treatment of stress-induced maladaptive behavior associated with depression. When compared to (R, S)-ketamine, FNM has been shown to exert rapid antidepressant actions with a low side-effect profile in mice by selectively antagonizing NMDA receptors [156]. Similar to memantine, FNM binds in a non-competitive manner and an open active state to the PCP site of the NMDA receptor as observed in radioligand binding studies ([^18^F]-FNM) [157]. FNM is currently in phase 1 clinical trials, offering hope for the treatment of neurodegenerative disorders, including post-traumatic stress disorder, AD, major depressive disorder and treatment-resistant depression [158]. Another promising rapid antidepressant agent is esmethadone (REL-1017), a dextro isomer of methadone with little or no activity towards the opioid receptor. Like memantine, esmethadone is a low-affinity non-competitive NMDA receptor antagonist. Esmethadone is currently in a phase 3 clinical trial for the treatment of major depressive disorder [159]. The development of these antagonists (Figure 9) emphasizes the prominent effect of excitotoxicity in neurodegenerative disorders and redefines polycyclic cage structures with NMDA receptor selectivity and fast kinetic interaction. Moreover, these polycyclic cages are permeable to the BBB with a minimal side-effect profile. This offers the opportunity to explore more polycyclic cages acting at the PCP binding site in an open active NMDA receptor state. Several structural-related polycyclic cages have been shown to display neuroprotective effects against glutamate-induced toxicity and other degenerative processes [160]. However, the reported findings are based solely on experimental data, and further exploration through clinical studies on these groups of NMDA receptor antagonists is warranted. 

The multifaceted nature of many neurodegenerative disorders makes designing and developing potential treatments complex and highly challenging. Factors contributing to the degenerative process are interrelated, including excitotoxicity, oxidative stress, neuroinflammation, protein aggregation, and mitochondria dysfunction. The majority of NMDA receptor antagonists are designed to target excitotoxicity, corresponding to a single-target approach. Moreover, a few of these antagonists fail to cross the BBB due to their hydrophilic nature. In recent years, the focus has been on designing and developing multifunctional agents with the potential to address glutamate-induced toxicity and other therapeutic targets, adopting a multi-target approach to drug development. These antagonists, sharing functional and structural similarities to amantadine and memantine, exhibit polycyclic cage structures that incur lipophilicity and enhanced permeation via the BBB. For example, a series of triazole-bridged aryl adamantane derivatives have been explored as multifunctional agents for the potential treatment of AD. These derivatives demonstrated potent inhibition of acetylcholinesterase enzymes, Aβ aggregation, and NMDA receptor, as well as good BBB permeability and a good safety profile in neuronal cell lines like SH-SY5Y neuroblastoma cells. This unique attribute makes them promising candidates for the treatment of AD [161]. Similarly, a series of polycyclic propargylamine and acetylene derivatives were investigated, revealing multifunctional activities, including neuroprotection, monoamine oxidase (MAO) inhibition, anti-apoptotic activities, and inhibition of NMDA receptors and voltage-gated calcium channels [162]. Another study explored some carbamate-based cholinesterase inhibitors, with structural similarities to acetylcholine, as potential multifunctional agents for AD treatments. These inhibitors exhibit diverse scaffolds, such as physostigmine, isosorbide, quinazoline, quinoline, xanthone, chalcone, flavonoid, indole-like, resveratrol and coumarin derivatives. In addition to their anti-cholinesterase activity, preclinical findings suggest multiple activities, including antioxidant properties, anti-neuroinflammation, metal chelation, neuroprotection, monoamine oxidase inhibition, neurotrophic effect and/or reduction in Aβ aggregation. Hence, they represent promising multifunctional candidates for the treatment of AD [163,164]. However, the majority of the available data for these inhibitors or antagonists stems from preclinical or experimental studies. There is a pressing need for in vivo and clinical studies to establish the therapeutical clinical efficacy of these groups of compounds.

Currently, only two FDA-approved multifunctional drugs (Namzaric^®^ and Auvelity^®^) are available, each containing an NMDA receptor antagonist and another therapeutic agent, for the treatment of neurodegenerative disorders. Namzaric^®^ is marketed for moderate to severe AD, while Auvelity^®^ is designated for the treatment of agitation associated with AD. Despite the enhanced activity and formulation adherence of each drug over its components, they have demonstrated similar side-effect profiles to their parent drugs [165,166,167]. Therefore, there is a need for a multifunctional hybrid or hybrids capable of antagonizing NMDA receptors, providing symptomatic relief, and targeting other degenerative processes such as neuroinflammation, oxidation and mitochondrial dysfunctions. Designing and developing selective GluN2B antagonists/modulators with polycyclic moieties and multitarget properties would be highly desirable. Such a multifaceted approach, with polycyclic scaffolds that incur good bioavailability, holds significant promise in addressing neurodegenerative disorders [168]. 

## 5. Conclusions

The significance of glutamate-induced excitotoxic death in the pathogenesis of neurodegenerative disorders is well established. With this knowledge, the ideal approach would be to use NMDA receptor antagonists to halt the degenerative process. Despite years of research, developing such agents has yielded little success, as current treatments, such as amantadine and memantine, only offer symptomatic relief. Many competitive and noncompetitive NMDA receptor antagonists have been explored but are marked by undesirable psychotomimetic side effects. These adverse effects are linked to the strong NMDA receptor-binding affinity or metamodulation of NMDA receptors by these antagonists that negatively influence their physiological functions. Interestingly, a more detailed exploration of the structure and function of NMDA receptors has led to the development of some selective negative allosteric modulators with good side-effect profiles. These modulators offer promise in addressing neurodegenerative disorders.

The distinctive biophysical features and localization of NMDA subunits provide a great opportunity for developing clinically effective drugs with optimum safety profiles. For instance, extrasynaptic neurons are rich in the GluN2B subunit and serve as key mediators of excitotoxic neuronal cell death. Therefore, selectively blocking this subunit would be therapeutically beneficial in addressing glutamate-induced cell death, especially in conditions such as AD and PD, where neurons in the brain cortex, hypothalamus, or striatum are predominantly affected. However, some of the developed antagonists fail in clinical trials due to poor bioavailability or lack of recruitment. One could explore the use of nano-drug formulations like solid lipid nanoparticles and nanostructured lipid carriers to improve the BBB permeability of these antagonists. Moreover, their selectivities toward the different GluN2 subtypes are often relative. This is majorly due to the amino acid sequence similarities existing among GluN2 subunits. For example, the overall amino acid sequence homology of GluN2A and GluN2B subunits are nearly identical, making it difficult to determine the strong subunit specificity of certain ligands [148,169]. Designing and developing agents that specifically and selectively target only GluN2B subunits is crucial for overcoming these challenges and improving the success rate in clinical trials [84]. This could be achieved by utilizing a receptor-based virtual screening method to identify amino acid residues that are unique to GluN2B subunits. Subsequently, a structure-based virtual screening technique would be employed to identify small molecules with optimum binding interactions with these targets. This targeted approach holds promise for the development of novel NMDA receptor antagonists that effectively address glutamate-induced excitotoxicity with minimal side effects. 

However, the challenge is the subunit diversity in the NMDA receptor channel complex, which is highly complex and not completely understood, despite decade-long x-ray crystallography studies on these subunits [148,170]. One of the most intriguing and challenging aspects of studies involving certain NMDA receptor antagonists is the discrepancy between preclinical and clinical findings, ultimately resulting in clinical trial failures. This may, in part, be linked to the differences in LBD residues that exist between human and animal (rodent) NMDA receptor subunits. Additionally, the expression of these subunits may differ at each development stage of the animal as observed in rodents, leading to translational failure [148]. This highlights our limited understanding of the NMDA receptor structure and function, particularly in humans, which remains poorly defined to date. Further research and refinement in drug design and understanding the molecular structure and functions of the NMDA receptor are crucial to fully unlock the therapeutic potential of this strategy in treating neurodegenerative disorders. Elucidating the structure of human NMDA receptor subunits will be a step in the right direction. However, it requires the collaborative efforts of medicinal chemists, physicists, bioinformaticists, computational chemistry, and structural biology. 

## Figures and Tables

**Figure 1 pharmaceuticals-17-00639-f001:**
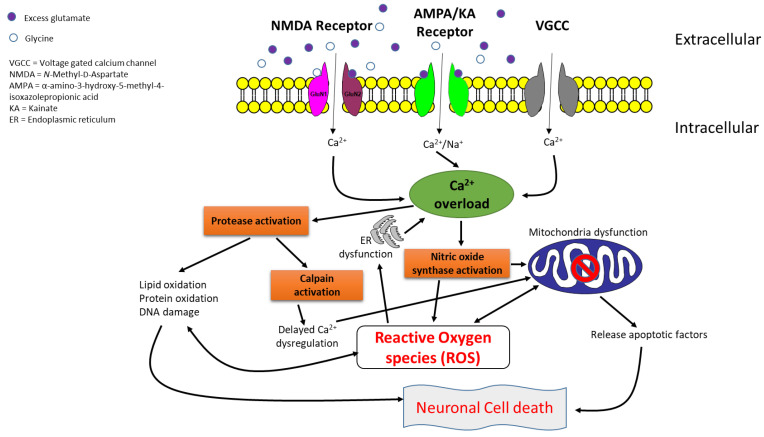
Neurodegenerative process associated with glutamate-induced excitotoxic cell death.

**Figure 2 pharmaceuticals-17-00639-f002:**
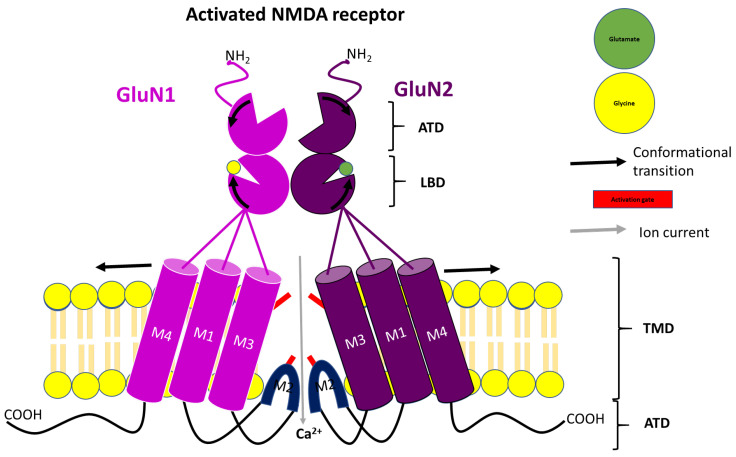
Schematic illustration of NMDA receptor activation.

**Figure 3 pharmaceuticals-17-00639-f003:**
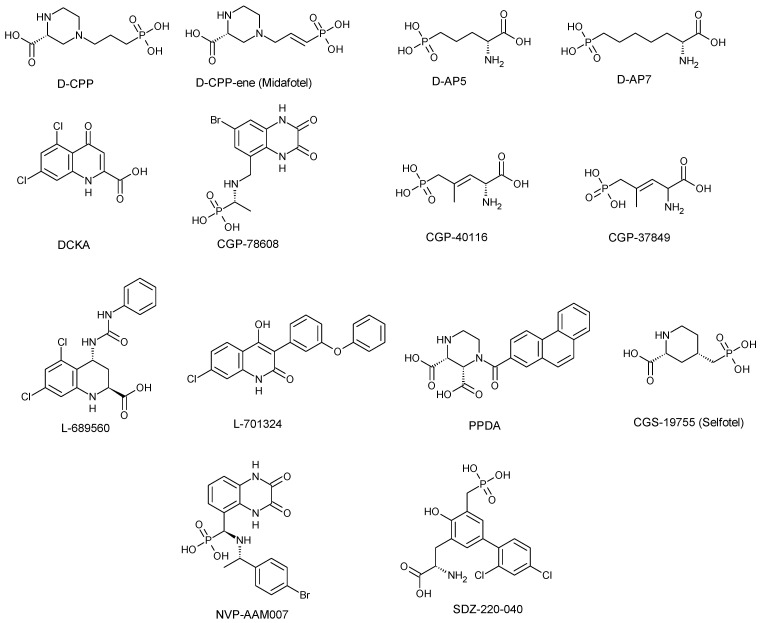
Structures of selected competitive NMDA receptor antagonists.

**Figure 4 pharmaceuticals-17-00639-f004:**
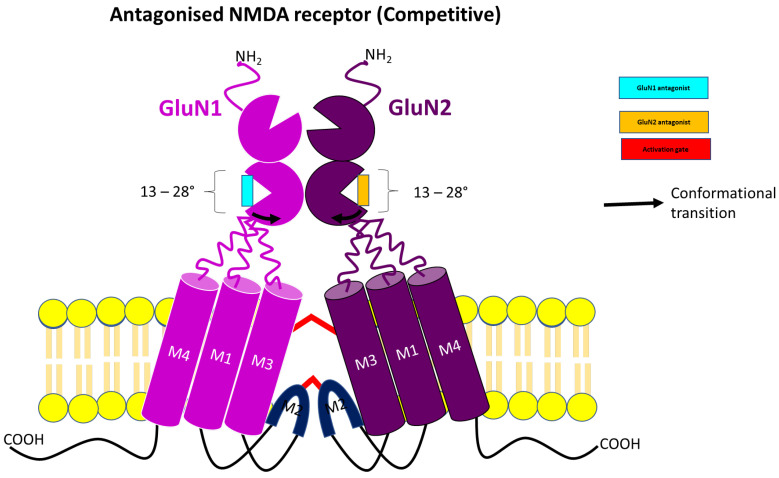
Schematic representation of competitive antagonism at NMDA receptor.

**Figure 5 pharmaceuticals-17-00639-f005:**
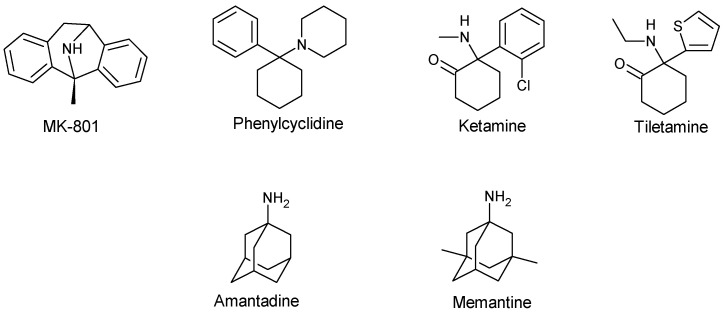
Structures of channel blockers or non-competitive NMDA receptor antagonists.

**Figure 6 pharmaceuticals-17-00639-f006:**
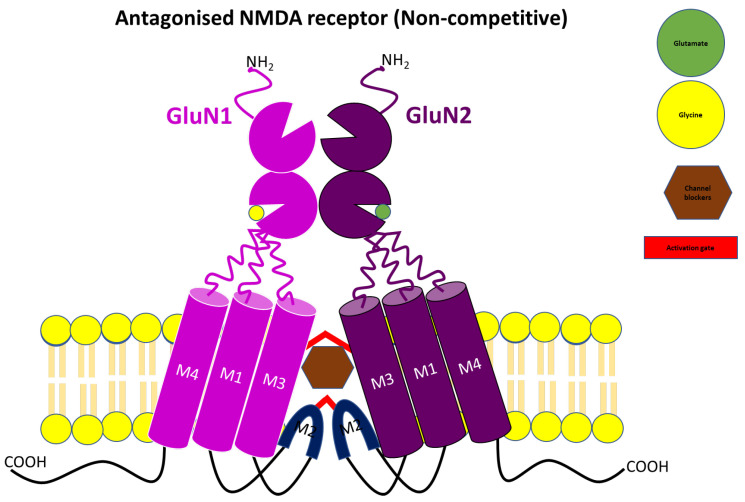
Schematic representation of NMDA receptor antagonism by channel blockers or non-competitive antagonists.

**Figure 7 pharmaceuticals-17-00639-f007:**
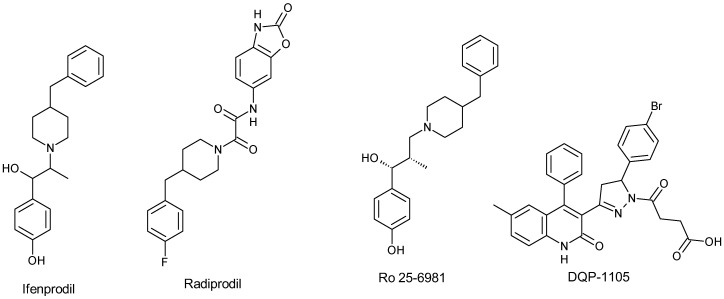
Negative allosteric modulators of NMDA receptors.

**Figure 8 pharmaceuticals-17-00639-f008:**
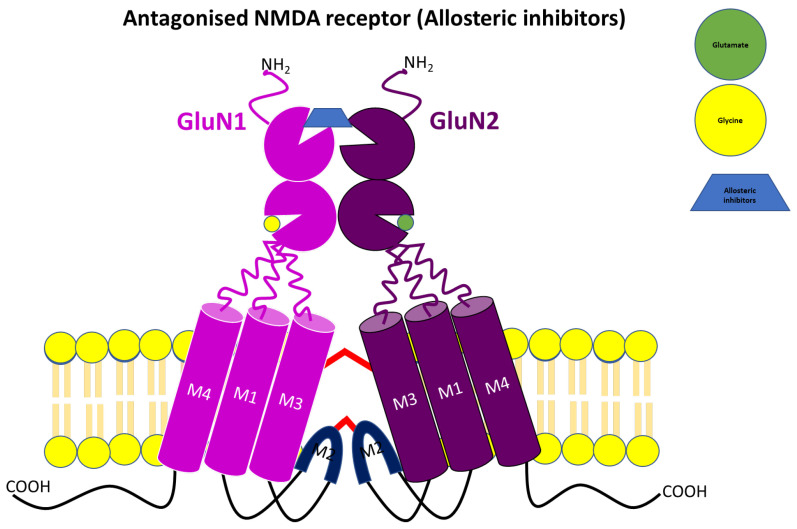
Illustration of negative allosteric modulators at NMDA receptor binding site.

**Figure 9 pharmaceuticals-17-00639-f009:**
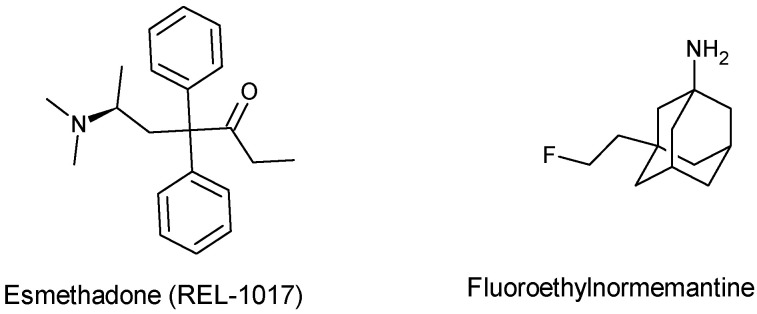
Non-competitive NMDA receptor antagonist presently in clinical trials.

## Data Availability

No new data were created or analyzed in this study. Data sharing is not applicable to this article.

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
