# Peer review of "NMDA Receptor Antagonists: Emerging Insights into Molecular Mechanisms and Clinical Applications in Neurological Disorders"

_pharmaceuticals, 2024, doi:10.3390/ph17050639_

Round 1
Reviewer 1 Report
Comments and Suggestions for Authors
The paper discusses chronic conditions like Alzheimer’s and Parkinson’s, characterized by progressive neuronal loss leading to cognitive and motor impairments. The global prevalence of these disorders is increasing, posing significant economic and social burdens. A key pathological process in neurodegenerative disorders is excitotoxicity, primarily mediated by glutamate. NMDA receptors play a central role in this process, and targeting them holds therapeutic potential, despite challenges like blood-brain barrier penetration and adverse effects. The review explores the molecular mechanisms of NMDA receptor antagonists, their structure, function, and types, as well as the challenges and future prospects in treating neurodegenerative disorders. It emphasizes the need for multifunctional agents targeting multiple degenerative processes and the importance of advancements in drug design and collaborative efforts to realize the therapeutic potential of NMDA receptor modulation. Specific comments:
1. The abstract provides a good overview of the paper’s content, focusing on NMDA receptor antagonists and their role in neurodegenerative disorders. However, it could benefit from a brief mention of the key findings or conclusions drawn from the review.
2. The paper discusses excitotoxicity and its implications in neurodegenerative disorders. It would be helpful to include a figure illustrating the excitotoxic cascade for better understanding.
3. The challenge of blood-brain barrier penetration by NMDA receptor antagonists is mentioned. Could the authors elaborate on the strategies being explored to overcome this obstacle?
4. The paper refers to clinical trials of selective GluN2B antagonists like ifenprodil and radiprodil facing obstacles. Can the authors provide more details on these trials and the specific challenges encountered?
5. Recent developments in phenanthroic and naphthoic acid derivatives are mentioned. Could the authors discuss the pharmacokinetic properties of these compounds as well?
6. The review mentions the adverse effects of NMDA receptor antagonists. It would be beneficial to have a table summarizing these effects alongside the compounds discussed.
7. The paper discusses the difficulty in targeting specific GluN2 subunits due to amino acid sequence similarities. Can the authors suggest potential molecular targets or strategies to achieve better specificity?
8. The use of nano-drug formulations is suggested as a means to improve bioavailability and BBB permeability. Can the authors provide examples of current research or trials utilizing these formulations?
9. The concept of multifunctional agents targeting multiple degenerative processes is intriguing. Could the authors provide examples of such agents currently under investigation?
10. The conclusion emphasizes the need for collaborative efforts in drug design. Can the authors comment on any existing collaborations or consortia focused on NMDA receptor modulation in neurodegenerative disorders?
Author Response
Please see attachment below

Reviewer 2 Report
Comments and Suggestions for Authors
The manuscript is titled NMDA Receptor Antagonists: Emerging Insights into Molecular Mechanisms and Clinical Applications in Neurological Disorders, by Ayodeji O. Egunlusi. Although the manuscript is in progress, further development are needed before it is suitable for publication. Currently, the manuscruipt would require more structure and innovation.
1. Major issues
1. The review structure is not easy to find.
2. The reason the authors want to cover this topic is really missing:
a) The authors should consider the essential aspects that could introduce the reader to the topic (e.g., neuodengative diseases); b) NMDA biological function with figures
c) Why do we need to know about the antigonoists?
d) What are the different types of antigonoists?
e) What is the basis for their classification?
f) How are they involved in the area of neurodegenerative diseases?
g) etc..
The the review is not about gathering information; it is more about discussing and interpreting it in a way that opens up the door to future development in the field. This might be currnetly lacking.
Comments on the Quality of English LanguageMinor English editing is needed.
Author Response
Please the attachment

Reviewer 3 Report
Comments and Suggestions for Authors
This article shows the implications of NMDA Receptor Antagonists into Molecular Mechanisms and Clinical Applications in Neurological Disorders. The topic is relevant, but the major deficiencies identified in both content and form need to be addressed based on the specific recommendations below:
1. Authors should check and apply the instructions for authors present in the template provided by the journal regarding author data, font, bibliographic style because there are currently big differences etc.
2. The aim of the paper must be presented separately in the last paragraph of the introduction and needs to be addressed from the perspective of describing the contribution to the field under analysis and the elements of scientific novelty presented.
3. The introduction should provide more data on pathophysiological mechanisms and less on epidemiological data which are not the subject of this paper. I suggest checking and referring to: PMID: 33446243 and PMID: 33215389
4. Paragraphs should start with indentation and figures should be more structured (figure with chemical structures is not observable).
5. In order to add value to the scientific literature, this review should present in tabular form studies that are being done/have been done on NMDA receptor antagonists with the most important pharmacological, efficacy and safety data through computational, cell culture, animal and clinical models.
6. Conclusions should be made strictly on the basis of the authors' observations and not by reference to other works, and detailed references to future research studies should be added.
Round 2
Reviewer 2 Report
Comments and Suggestions for Authors
Thank you for addressing my comments.
Kind regards
Comments on the Quality of English LanguageMinor corrections of English need to be done